# Expression-Based Diagnosis, Treatment Selection, and Drug Development for Breast Cancer

**DOI:** 10.3390/ijms241310561

**Published:** 2023-06-23

**Authors:** Qing Ye, Jiajia Wang, Barbara Ducatman, Rebecca A. Raese, Jillian L. Rogers, Ying-Wooi Wan, Chunlin Dong, Lindsay Padden, Elena N. Pugacheva, Yong Qian, Nancy Lan Guo

**Affiliations:** 1West Virginia University Cancer Institute/Mary Babb Randolph Cancer Center, West Virginia University, Morgantown, WV 26506, USA; qiye@mix.wvu.edu (Q.Y.); abcjia@gmail.com (J.W.); rebecca.raese@gmail.com (R.A.R.); jroger24@mix.wvu.edu (J.L.R.); yingwoow@bcm.edu (Y.-W.W.); lindadong2004@yahoo.com (C.D.); padden_lm@wvwc.edu (L.P.); epugacheva@hsc.wvu.edu (E.N.P.); 2Department of Pathology, West Virginia University, Morgantown, WV 26506, USA; bducatman@hsc.wvu.edu; 3Department of Biochemistry and Molecular Medicine, School of Medicine, West Virginia University, Morgantown, WV 26506, USA; 4Department of Radiation Oncology, School of Medicine, West Virginia University, Morgantown, WV 26506, USA; 5Pathology and Physiology Research Branch, National Institute for Occupational Safety and Health, Morgantown, WV 26505, USA; yaq2@cdc.gov; 6Department of Occupational and Environmental Health Sciences, School of Public Health, West Virginia University, Morgantown, WV 26506, USA

**Keywords:** atypical ductal hyperplasia (ADH), atypical ductal hyperplasia with cancer (ADHC), diagnosis, CRISPR-Cas9/RNAi, immunohistochemistry, triple-negative breast cancer (TNBC)

## Abstract

There is currently no gene expression assay that can assess if premalignant lesions will develop into invasive breast cancer. This study sought to identify biomarkers for selecting patients with a high potential for developing invasive carcinoma in the breast with normal histology, benign lesions, or premalignant lesions. A set of 26-gene mRNA expression profiles were used to identify invasive ductal carcinomas from histologically normal tissue and benign lesions and to select those with a higher potential for future cancer development (ADHC) in the breast associated with atypical ductal hyperplasia (ADH). The expression-defined model achieved an overall accuracy of 94.05% (AUC = 0.96) in classifying invasive ductal carcinomas from histologically normal tissue and benign lesions (*n* = 185). This gene signature classified cancer development in ADH tissues with an overall accuracy of 100% (*n* = 8). The mRNA expression patterns of these 26 genes were validated using RT-PCR analyses of independent tissue samples (*n* = 77) and blood samples (*n* = 48). The protein expression of PBX2 and RAD52 assessed with immunohistochemistry were prognostic of breast cancer survival outcomes. This signature provided significant prognostic stratification in The Cancer Genome Atlas breast cancer patients (*n* = 1100), as well as basal-like and luminal A subtypes, and was associated with distinct immune infiltration and activities. The mRNA and protein expression of the 26 genes was associated with sensitivity or resistance to 18 NCCN-recommended drugs for treating breast cancer. Eleven genes had significant proliferative potential in CRISPR-Cas9/RNAi screening. Based on this gene expression signature, the VEGFR inhibitor ZM-306416 was discovered as a new drug for treating breast cancer.

## 1. Introduction

Breast cancer is the most common cancer in women worldwide and has the second highest incidence rate in women in the United States after skin cancer. About 81% of breast cancer cases are invasive, and the remaining cases are considered benign, such as ductal carcinoma in situ (DCIS) and lobular carcinoma in situ (LCIS). Breast cancer is diagnosed with breast ultrasound, mammogram, magnetic resonance imaging (MRI), or biopsy. Due to these screening tools and improved early detection, the survival rate of breast cancer has increased over the years, with an average 5-year survival rate of 91% and a 10-year survival rate of 84% in women with non-metastatic invasive breast cancer [1].

Recent advancement in gene expression-based prognosis of breast cancer has also contributed to improved breast cancer treatment and overall survival. Oncotype DX [2] and MammaPrint [3,4] are available for ER-positive early-stage patients. The cost of Oncotype DX is covered by major healthcare insurance companies and the Medicare/Medicaid program in the US. After Oncotype DX was made commercially available to clinics, it entered a prospective clinical trial that finalized the patient stratification schemes in 2018 [5,6]. Breast cancer subtypes based on gene expression profiling [7,8,9], PAM50 (prediction of microarray using 50 classifier genes plus 5 reference genes) [10], the Nottingham Prognostic Index Plus (NPI+) [11], the Breast Cancer Index [12], The Rotterdam Signature [13], and EndoPredict [14] were also developed to aid treatment selection for invasive breast cancers. Other gene signatures for breast cancer prognosis were identified based on wound-healing response [15], immune response [16], and stromal gene expression [11].

Among the breast cancer subtypes, triple-negative breast cancer (TNBC, and basal-like) does not express hormone receptors and, thus, has limited therapeutic options [17]. TNBC accounts for 10–20% of breast cancer cases. Due to the high number of tumor-infiltrating lymphocytes (TILs), TNBC is considered the most immunogenic breast cancer subtype [18]. Anti-PD-L1 (atezolizumab) combined with nab-paclitaxel induced survival benefits in locally advanced or metastatic TNBC patients in a randomized phase III clinical trial [19]. Nevertheless, a significant portion of TNBC patients do not respond to immunotherapy. Tumor mutation burdens and PD-L1 expression are not associated with response to immune checkpoint inhibitors in breast cancer [20,21]. Novel biomarkers and therapeutic options are needed to improve TNBC treatment outcomes.

Despite the novel discoveries and successful clinical applications of molecular biomarkers for breast cancer prognosis, there are currently no clinically applied multi-gene assays for early breast cancer detection. More than 80% of breast cancer cases are discovered when a lump in the breast is detected with the fingertips [22]. Current breast cancer screening tools such as mammograms are controversial in terms of benefits versus harm for use in routine testing [23,24]. The medium mammogram-detectable tumor size was reported as 7.5 mm [25]. False results could arise due to technical issues or a tumor size smaller than that. Molecular assays for the intraoperative evaluation of sentinel nodes have been developed using cytokeratin 19 (*CK19*) mRNA amplification [26] and mammaglobin (MG) and CK19 immunohistochemistry (IHC) [26], both achieving high specificity and sensitivity. The molecular diagnosis of triple-negative breast cancer is primarily based on the IHC of the estrogen receptor (ER), progesterone receptor (PR), and human epidermal growth factor receptor 2 (HER2) [27]. EGFR/HER2 assessment is used in breast molecular diagnosis and therapy [28]. Specific translocations characterizing some special types of mammary carcinomas could potentially be used as diagnostic companion tests [29]. A balanced translocation between chromosomes 12 and 15 creating a new *ETV6-NTRK3* fusion gene is a primary event in human secretory breast carcinoma [30]. Adenoid cystic carcinoma of the breast, a rare histological type of triple-negative breast cancer with indolent clinical behavior, is characterized by the *MYB-NFIB* fusion gene [31]. Approximately 5–10% of breast cancer incidents are deemed inheritable [32]. Genetic tests based on *BRCA1* and *BRCA2* are clinically available to estimate hereditary breast–ovarian cancer familial risks, which account for about 90% of inheritable cases. Other gene mutations were linked to a substantial minority of inheritable breast cancers, including *TP53, PTEN, STK11, CHEK2, ATM, BRIP1*, and *PALB2* [32]. For the majority of the breast cancer patient population, there are currently no clinically applicable gene tests to detect a precancerous state in histologically normal tissue and benign lesions or to predict which premalignant lesions will develop into invasive breast cancer. Specifically, biomarkers found in biological fluids, blood in particular, are the most promising for the fast development of screening assays for early detection of breast cancer [33].

Emerging biomarkers for breast cancer diagnosis should provide additional information relevant to prognosis and the selection of therapy [27]. We previously identified and validated a 28-gene prognostic signature in more than 2000 breast cancer patients in both early and late stages of any ER or lymph node status at the time of diagnosis [34,35]. It is also prognostic of the clinical outcome in multiple epithelial cancers, including ovarian cancer [35,36]. With the evidence of its prognostic capacity established in our previous studies [34,35,36], this study sought to investigate (1) its potential clinical utility in breast cancer diagnosis in solid tissues and blood samples that we collected; (2 its ability to further refine prognosis within TNBC using The Cancer Genome Atlas (TCGA); (3) its association with immune infiltration and immune cell type activities in TCGA breast cancer patient tumors; (4) its proliferation potential in CRISPR-Cas9/RNAi screening in human breast cancer cell lines in the public Cancer Cell Line Encyclopedia (CCLE); (5) its association with 18 National Comprehensive Cancer Network (NCCN)-recommended drugs for treating breast cancer in the CCLE panel. Finally, based on this 28-gene expression signature, new drug options for treating breast cancer were identified using Connectivity Map (CMap) [37,38].

## 2. Results

### 2.1. 28-Gene Signature as a Diagnostic Assay for Breast Cancer

We studied the diagnostic performance of the 28-gene signature in two independent patient cohorts. In both cohorts, the M5P algorithm was used to construct a classifier with the signature genes to identify invasive breast cancer from normal, benign, or premalignant tissues. The final prediction performance was computed from the leave-one-out cross-validation.

In the first cohort from Chen et al. [39] (*n* = 185), the constructed classifier was highly precise in discriminating normal breast tissues from breast cancer tumors, with an area under the ROC curve (AUC) of 0.96 (Figure 1A). Specifically, using the expression profile of the 26 available genes as predictors, the malignant scores estimated by the classifier correctly identified normal breast tissue samples from invasive ductal carcinoma samples 94% of the time (Figure 1B). In the second cohort of patients with atypical ductal hyperplasia (ADH) from Poola et al. [40] (*n* = 8), using the 25 available genes, the M5P classifier perfectly discriminated between patients with or without invasive breast cancer (AUC = 1, accuracy = 100%; Figure 1C,D). These results demonstrate that the 28-gene signature is a strong classifier of breast cancer tumors versus benign tissues, with potential implications in breast cancer diagnosis.

### 2.2. RT-PCR Validation of 26 Genes out of the 28-Gene Signature in Breast Cancer Tissue Samples

Microfluidic low-density arrays were designed to quantify the expression of 26 known genes in the gene signature in an independent patient cohort (Appendix A). Using the mRNA expression profiles of the 26 marker genes, 9 out of 12 normal breast tissue samples in the third patient cohort were separated from 65 breast cancer tumors in hierarchical clustering analysis (Figure 2A), indicating its diagnostic capacity to classify breast cancer tumors from normal breast tissues in RT-PCR assays. Fourteen genes had significant (*p* < 0.05) differential expression in breast cancer tumors vs. normal breast tissues. Specifically, *ACOT4, DDOST, IRF5, MAP2K2, MCM2, PBX2, S100P*, and *SSBP1* had significant (*p* < 0.05) upregulation in invasive carcinoma compared with normal breast tissues (Figure 2B). *FGF2* and *IGHA1* had significant (*p* < 0.05) downregulation in invasive carcinoma compared with normal breast tissues. *FGF2* was also downregulated (*p* < 0.05) in DCIS vs. normal breast tissues (Figure 2B). Multiple genes were downregulated (*p* < 0.05) in sarcoma tumors, including *HSP90AB1, IGHA1, INSM1, IRF5, MAP2K2*, and *SEH1L* (Figure 2B). The RT-PCR results are provided in Appendix A. The results of the analyses of each gene by cancer stage and by tumor grade are provided in Appendix A. The protein expression of FGF-2, RAD52, RAD50, PBX2, MAP2K2 (MEK2), and S100P was also validated with immunohistochemistry in invasive breast cancer tumor tissues. RAD52 and PBX2 had differential protein expression among breast cancer stages (Appendix A). 

### 2.3. RT-PCR Validation of 28-Gene Signature in Breast Cancer Blood Samples

The mRNA expression of the 28-gene signature was further examined in breast cancer blood samples of the fifth patient cohort (*n* = 48) using microfluidic low-density arrays in RT-PCR. Seven genes showed significant (*p* < 0.05) differential expression in DCIS and/or invasive breast cancer patient blood samples compared with blood samples from healthy individuals (Figure 3A, enrollment criteria provided in Appendix A). *FGF2* and *S100P* were overexpressed (*p* < 0.05) in both DCIS and invasive breast cancer patient blood samples. *IRF5*, *MAP2K2 (MEK2)*, and *ZBTB7B* were significantly overexpressed (*p* < 0.05) in invasive breast cancer blood samples but not in DCIS blood samples. *MCM2* and *TXNRD1* were underexpressed (*p* < 0.05) in DCIS patient samples but not in invasive breast cancer samples (Figure 3A). *IRF5, MAP2K2 (MEK2),* and *S100P* had concordant overexpressed mRNA in invasive breast cancer tissues and blood samples. Using the mRNA expression of these seven genes, normal blood samples were separated from breast cancer patient blood samples in hierarchical clustering analysis (Figure 3B). The protein expression of *RAD52, RAD50, PBX2, MAP2K2 (MEK2),* and *S100P* was also validated with immunohistochemistry in invasive breast cancer tumor tissues (Appendix A). We acknowledge that there were only seven normal blood samples. This was a feasibility study that warrants a larger-scale analysis to substantiate its potential implications in the diagnosis of breast cancer in the future.

### 2.4. Prognosis within Breast Cancer Subtypes Using Next-Generation Sequencing Data

The Cancer Genome Atlas on breast cancer (TCGA-BRCA) dataset was randomly split into a training (*n* = 547; 531 patients had sufficient survival information) and a testing set (*n* = 548; 520 patients had sufficient survival information) to develop and validate the prognostic model. Using the 25 available genes (Appendix A), a multivariate Cox model was constructed to compute the risk score for each patient. The same model was then applied to the testing set, employing identical gene coefficients and the cutoff for patient stratification. Kaplan–Meier analysis revealed that patients with a risk score less than or equal to 7.59 had significantly longer survival than patients with a risk score greater than 7.59 in both the training set (*p* = 5.14 × 10^−6^, HR: 2.607 [1.7, 3.997]; Figure 4A) and the testing set (*p* = 0.016, HR: 1.847 [1.114, 3.061]; Figure 4B). The details of the training and testing data and the model coefficients are provided in Appendix A.

Next, we evaluated the prognostic value of our model by performing survival analysis on four PAM50 subtypes in TCGA-BRCA patients. These subtypes include basal-like (*n* = 140), luminal A (*n* = 419), luminal B (*n* = 180), and Her2-positive (*n* = 62). Kaplan–Meier analysis demonstrated that the low-risk patient group had significantly better survival outcomes than the high-risk patient group in the basal-like (Figure 4C) and luminal A subtypes (Figure 4E). Specifically, the 6-year survival rates were 93.09% vs. 58.57% for low-risk vs. high-risk basal-like patients, respectively, and the 6-year survival rates were 86.88% vs. 67.9% for low-risk vs. high-risk luminal A patients, respectively. These results indicated that our prognostic model can further refine the prognosis within the basal-like and luminal A subtypes. In contrast, our prognostic model cannot further stratify the luminal B or Her2-positive subtypes (Appendix A). In the random split, we tried to keep the same proportion of samples in the training set (*n* = 547) and the testing set (*n* = 548). However, the specific number of each PAM50 subtype in the training set and the testing set will vary slightly in the random split. In the TCGA-BRCA cohort, luminal A has the largest number of patients (419 out of 1051) in the survival analysis. This might be the reason our classifier produced similar results for luminal A (Figure 4E) and the testing set in the first 10 years after surgery (Figure 4B).

For each patient sample in the TCGA-BRCA cohort, we computed the xCell [41] scores, representing the transcriptional activities of all immune, epithelial, and stromal cell types using the corresponding RNA sequencing data. We identified immune cell types with significantly different activities in the low-risk vs. high-risk groups for the basal-like (Figure 4D) and luminal A (Figure 4F) subtypes. Cell type activity changes more in high-risk patients when the log_2_ ratio is positive, whereas it varies more in low-risk patients when the log_2_ ratio is negative. The following cell types were more active in basal-like patients with a more favorable survival outcome: Tgd (gamma delta T) cells, erythrocytes, CD4^+^ T cells, NK cells, neutrophils, memory B cells, CD4^+^ naïve T cells, megakaryocytes, eosinophils, CD8^+^ Tem (effector memory cells), CD4^+^ memory T cells, CD4^+^ Tem, plasmacytoid dendritic cells (pDC), granulocyte–macrophage progenitor (GMP), myocytes, dendritic cells (DC), plasma cells, class-switched memory B-cells, and ImmuneScore. The cell types that were more active in basal-like patients with a poor prognosis included the following: mesangial cells, NKT, keratinocytes, basophils, hepatocytes, mv endothelial cells, astrocytes, epithelial cells, CD4^+^ Tcm (central memory cells), platelets, ly endothelial cells, and sebocytes. In luminal A patients with a good prognosis, the following cell types had more variation in transcriptional activities: mv endothelial cells, macrophages, immature dendritic cells (iDC), fibroblasts, pericytes, endothelial cells, StromalScore, common lymphoid progenitor (CLP), MicroenvironmentScore, megakaryocytes, and epithelial cells. CD4^+^ Tcm, CD8^+^ naïve T cells, CD4^+^ memory T cells, sebocytes, class-switched memory B-cells, and mesangial cells had more varied activities in luminal A patients with a poor prognosis.

### 2.5. Correlation of Signature Genes with Immune Infiltration

We assessed the correlation between immune infiltration and the expression of 25 genes (Figure 5; *IGHA1* is not available, and *FAT* is named *FAT1*) in TCGA-BRCA patients (*n =* 1100) using Tumor Immune Estimation Resource (TIMER) 2.0 [42,43]. The results revealed the immune infiltration of B cells, macrophages, myeloid dendritic cells, neutrophils, CD4^+^ T cells, and CD8^+^ T cells. The majority of our signature genes showed a significant correlation with immune infiltration in at least one of the examined cell types in luminal A patients or breast cancer patients with all subtypes combined (Figure 5).

### 2.6. Protein Expression Validation of the Signature Genes

Due to the limited sample size in our RT-PCR analysis, we correlated the RT-PCR results with SEER breast cancer patient data (Appendix A). This methodology correlates gene expression in a clinical cohort with large-scale SEER data based on a combined index of tumor grade and T, N, M in the cancer stage, as described in our previously published article [44]. This process can extrapolate gene expression into a larger patient population and examine the association with treatment outcomes. In this analysis, *RAD52* and *SMARCD2* had significant associations with patient outcomes who received surgery only, radiation, or surgery plus radiation. Detailed results are shown in Appendix A.

Based on these results, the protein expression of the signature genes was further validated using Western blots of cell lines and immunohistochemistry assays of tumor tissues in the fourth patient cohort to substantiate the functional involvement of the marker genes in breast cancer tumors. In Western blots, RAD50 was overexpressed in breast cancer cells (MCF7, AU565, and BT474) compared with normal breast epithelial cells (MCF10A; Figure 6A). RAD50 was not expressed in TNBC cells MDA453. PBX2 was underexpressed in luminal A breast cancer MCF7 and TNBC MDA453 cells but was overexpressed in Her2-positive AU565 and BT474 cells [45], compared with normal breast MCF10A cells (Figure 6A). The RAD52 protein was overexpressed in breast cancer cells (MCF7, AU565, BT474, and MDA453) compared with the immortalized breast cells (MCF10A; Figure 6A), and its mRNA expression was not significantly different in breast cancer tumors compared with normal breast tissues (Figure 2B). Both RAD50 and RAD52 are functionally involved in DNA repair [46], and their protein expression was largely correlated with the focal adhesion protein NEDD9 in breast cancer cells (Figure 6A). NEDD9 (HEF1) plays an important role in cell attachment and division in mitosis [47], and promotes breast cancer development [48] and colorectal cancer progression [49]. The protein expression of RAD50, RAD52 (Figure 6B), and PBX2 (Figure 6C) was analyzed in breast cancer tumors using immunohistochemistry (IHC). Detailed information on the protein expression level distributions of RAD50, RAD52, and PBX2 in IHC is provided in Appendix A.

We performed further survival analyses based on the IHC results. The Kaplan–Meier results showed that the patients with higher IHC scores in RAD52 and/or PBX2 survived for a significantly shorter time than other patients (Figure 7). SMARCD2 protein expression assessed with IHC was not prognostic in the examined breast cancer cohort (Appendix A).

In the TCGA-BRCA patient cohort, the cutoffs of RAD52 and PBX2 mRNA expression were determined with the most significant stratification (with the lowest *p*-value) in survival analysis. According to the Kaplan–Meier analysis of the TCGA-BRCA data, patients with a *RAD52* mRNA expression level lower than 6.485 had a better prognosis for survival than those with a *RAD52* mRNA expression level higher than 6.485 (Figure 8A). Although the survival analysis result was not significant, the two groups had a good separation after 5 years. These results are consistent with the RT-PCR results in our patient cohort with the SEER breast cancer patients that higher *RAD52* mRNA expression was associated with poor patient outcomes in those who received surgery only, radiation, or surgery and radiation (Appendix A). The mRNA and protein expression patterns of RAD52 were consistent. Unlike PBX2 protein expression, the patients with a *PBX2* mRNA expression level higher than 9.93 had a better survival outcome than the patients with a *PBX2* mRNA expression level lower than 9.93 in the TCGA-BRCA data (Figure 8C). Based on the mRNA expression level of *RAD52* and *PBX2* together, the TCGA-BRCA patients were divided into four groups: both low (*RAD52* < 6.485 and *PBX2* < 9.93), both high (*RAD52* > 6.485 and *PBX2* > 9.93), *RAD52* low—*PBX2* high (*RAD52* < 6.485 and *PBX2* > 9.93), *RAD52* high—*PBX2* low (*RAD52* > 6.485 and *PBX2* < 9.93). The patients in the “both low” group had the best survival outcome among all four groups according to the Kaplan–Meier analysis results, which was consistent with their protein expression in the survival analyses (Figure 7E,F). The patients in the “*RAD52* high—*PBX2* low” group survived for the shortest length of time among the four groups (Figure 8E).

*RAD52* is functionally important in DNA repair, cancer susceptibility, and immunodeficiency [50]. *PBX2,* a homeobox gene, is 92% identical to human proto-oncogene *PBX1* and is widely expressed in different states of differentiation and development [51]. The *Prep1*/*PBX2* complex regulates *CCL2* expression, which is associated with numerous inflammatory diseases including HIV [52]. We sought to identify immune cell types that significantly (two-sample *t*-tests; *p* < 0.05) differed in activity across breast cancer patient groups with high and low expression of *RAD52* and *PBX2* linked to their survival outcomes (Figure 8A,C). The log_2_ ratios of xCell scores for tumors with high vs. low expression of *RAD52* and *PBX2* are shown in Figure 8B,D in the prognostic stratifications, respectively. For the four patient groups, a positive log_2_ xCell score ratio of high- vs. low-expression groups showed that cell type activities varied more in high-expression tumors; a negative log_2_ xCell score ratio of high- vs. low-expression groups showed that cell type activities varied more in low-expression tumors. The xCell scores were also computed for the four-group patient stratifications using combined *RAD52* and *PBX2* expression levels (Figure 8E). Immune cell types with a significant difference in activities (ANOVA tests; *F*-statistics, *p* < 0.05) among the four patient groups were identified (Figure 8F). Detailed statistical test results, including pair-wise comparisons in post hoc Tukey’s HSD tests following ANOVA tests, are available in Appendix A.

### 2.7. Proliferation Potential of the Signature Genes

The functional role of the 28 signature genes in breast cancer cell proliferation was evaluated in publicly available high-throughput CRISPR-Cas9 (*n =* 48) and RNAi (*n =* 34) screening data for human BRCA cell lines. Available genes with a significant dependency score (<–0.5) in the BRCA cell lines are shown in Figure 9, indicating that a knockdown/knockout of these genes significantly affects BRCA tumor cell growth in vitro. *SSBP1*, *TXNRD1*, *SLC25A5*, *DDOST*, *SEH1L*, and *MCM2* had a significant effect in at least 50% of breast cancer cell lines in the CRISPR-Cas9 and/or RNAi screening data.

### 2.8. Association with Drug Response

We found 18 NCCN-recommended regimens for preoperative/adjuvant/systemic/targeted treatment for breast cancer in the CCLE drug screening data. By utilizing the CCLE mRNA and proteomics profiles in human BRCA cell lines, we identified genes sensitive and resistant to these 18 drugs among 26 available signature genes (Table 1). Specifically, we classified genes as sensitive or resistant depending on whether they were significantly (*p* < 0.05; two-sample *t*-tests) overexpressed in sensitive BRCA cell lines or resistant BRCA cell lines for a given drug. Pan-sensitive genes were defined as those that were sensitive to any of the 18 medicines under study and were not resistant to any of them. Likewise, the genes that were determined to be resistant to any of the 18 medicines and not sensitive to any of them were labeled as pan-resistant genes. *HSP90AB1*, *INPPL1*, *IRF5*, and *RAD50* were pan-resistant genes. *PLSCR1* and *PBX2* were pan-sensitive genes at the mRNA expression level. FGF2 was resistant to alpelisib, and S100P was sensitive to olaparib in both mRNA and protein data.

Data from the Profiling Relative Inhibition Simultaneously in Mixtures (PRISM) and Genomics of Drug Sensitivity in Cancer (GDSC1/2) drug screening programs were included in our research. The information on the selected genes’ fusions and mutations in human BRCA cell lines is included in Appendix A.

### 2.9. Discovery of New Drugs with CMap

After substantiating the associations with patient survival, drug response, and proliferation of our 28-gene signature, we sought to identify new drugs for treating breast cancer based on this gene expression signature. Among the 26 known signature genes, we defined downregulated genes as CMap input with the following criteria: (1) proliferation genes that had significant dependency scores in at least 50% of the tested BRCA cell lines in CRISPR-Cas9 (*n =* 48) or RNAi (*n =* 34); (2) survival hazard gene in TCGA-BRCA mRNA prognostic analysis (univariate Cox model *p* < 0.05 and hazard ratio > 1); (3) pan-resistant genes in mRNA expression data for the studied 18 drugs. The final downregulated genes were *MCM2, SEH1L, SSBP1, DDOST, SLC25A5, TXNRD1, RAD50, TOMM70A, HSP90AB1, INPPL1,* and *IRF5*. Detailed information was provided in Appendix A.

A total of 17 candidate new or repositioning medicines were found using CMap, along with the significantly enriched chemical sets (*p* < 0.05, connectivity score >0.9; Appendix A). To find out if these candidate compounds can effectively inhibit the development of breast cancer cells, the half-maximal inhibitory concentration (IC_50_) and the half-maximal effective concentration (EC_50_) values of the drugs in the PRISM data were investigated in the CCLE human BRCA cell lines (*n* = 22). Six drugs with small average measurement values may potentially inhibit the growth of BRCA cells with a safe dose. The average IC_50_ and EC_50_ values of these six drugs in the PRISM screening data are shown in Figure 10.

For each discovered drug for treating breast cancer, we further identified resistant and sensitive genes associated with its drug response on a genome-wide scale. Dasatinib and PP2 had concordant sensitive/resistant results in mRNA and proteomics data, as shown in Table 2. We did not find sensitive/resistant genes with concordant mRNA and protein expression for the other four drugs. Non-silent mutations and fusions in human BRCA cell lines for the identified drug response genes were also revealed. Detailed information is provided in Appendix A.

## 3. Discussion

Breast cancer remains the most common cancer in women worldwide. Alongside inheritable genetic risk factors causing about 5–10% of breast cancer cases [32], the common modifiable risk factors include obesity, drinking alcoholic beverages [53], and smoking [54]. The Gail model is currently used to estimate breast cancer risk based on demographic information. One study cautions healthcare professionals in counseling individual patients with atypia using the Gail model because the results show that the Gail model significantly underestimates the risk of breast cancer in women with atypia, and its ability to classify women with atypia into those who do and do not develop breast cancer is limited [55]. The Gail model also underestimates genetic risks in breast cancer [56] and offers little improvement in breast cancer risk prediction in ER+ patients [57]. Biomarker-based models revealing intrinsic molecular mechanisms underlying breast cancer are needed to improve its diagnosis, early detection, and intervention.

The past two decades have witnessed paradigm shifts in medicine, with ground-breaking discoveries from genomic studies and successful translation into clinical practices and health insurance policies. In the field of breast cancer, two multi-gene assays, Oncotype DX and MammaPrint, are commercially available for the prognosis of early-stage ER+ patients with invasive breast cancer in the US and Europe. Such advancement demonstrates the promise of utilizing genomic biomarkers for precision medicine. Recent research has utilized “multi-omics” technology including genome-scale DNA copy number variation (CNV), DNA methylation, mRNA, microRNA (miRNA), exome sequencing, and reverse-phase protein arrays to reveal the landscape of breast cancer and other cancer types, such as in The Cancer Genome Atlas (TCGA). Integrated CNV and mRNA sequencing analysis has been used to identify novel subtypes of breast cancer [58]. Differential expression of miR10b, miR34a, miR155, and miR195 was found in sera of breast cancer patients and healthy control individuals using qPCR [59]. Nevertheless, there are no clinically applied gene tests for breast cancer diagnosis in the general population.

Initially motivated to discover a prognostic genomic signature for the general breast cancer patient population, including patients with all cancer stages and ER/nodal status, we identified a 28-gene prognostic signature from a population-based breast cancer cohort [34] and validated it in more than 2000 patients with all cancer stages and ER statuses [35]. The prognostic capacity of the 28-gene signature is beyond early-stage, ER+/lymph node-negative breast cancer. In this study, we showed that this gene signature can further refine prognosis within the TNBC and luminal A BRCA subtypes and revealed immune cell types with distinct activity profiles linked to different prognostic groups within these subtypes. The majority of the signature genes had a significant association with immune infiltration in TCGA-BRCA patient tumors. This 28-gene signature is also prognostic of clinical outcomes in multiple epithelial cancers, including ovarian cancer [36].

In this study, we sought to investigate if 26 known genes of this 28-gene signature are potentially also diagnostic of breast cancer and able to identify those with a high potential for developing invasive breast cancer in patients with atypical ductal hyperplasia (ADH). The overall accuracy of discriminating normal breast tissue from breast cancer tumors was 94% and the accuracy was 100% in classifying patients with ADH who did not develop future cancer and those who did (ADHC). This gene signature was also validated in RT-PCR in an independent breast cancer cohort for separating normal breast tissues from breast cancer tumors including DCIS, invasive carcinoma, and sarcoma. The protein expression of multiple signature genes was validated in breast cancer and normal breast cell lines using Western blots and tumor tissues with immunohistochemistry. Using seven differentially expressed signature genes in blood samples of breast cancer patients versus healthy women in RT-PCR, the normal individuals were again separated from breast cancer patients, including DCIS and invasive carcinoma. These results show that it is feasible to use the 28-gene signature for early diagnostic detection using patient tissues and blood samples. If this gene assay can be validated in prospective evaluation in a larger patient cohort, it will have the potential utility in clinics to (1) advise the need for biopsy to confirm breast cancer diagnosis after the initial screening through using this minimally invasive blood-based gene assay; (2) estimate the potential for developing malignancy in patients diagnosed with benign lesions of ADH; (3) predict the likelihood for tumor recurrence/metastasis and aid the selection of specific therapy in patients diagnosed with invasive breast carcinoma. The presented protein expression patterns in this panel of genes reveal important information on their involvement in breast cancer initiation and progression. These results warrant large-scale validation in independent patient cohorts for clinical applications.

Among the genes with concordant mRNA and protein expression in the validation cohorts, *MAP2K2 (MEK2), S100P*, and *PBX2* were all upregulated in terms of mRNA expression levels and had strong protein expression in invasive breast cancer tumors. *MEK2* controls the activation of the *MKK3/MKK6-p38* axis and has an essential impact on the MDA-MB-231 breast cancer cell survival and cyclin D1 expression [60]. Combined kinase inhibitors of MEK1/2 and either PI3K or PDGFR are efficacious in treating triple-negative breast cancer [61]. S100P by itself or together with Ezrin promotes the trans-endothelial migration of TNBC cells [62]. Consistent with our findings [34,35], other groups also reported the association of S100P mRNA [63] and protein expression [62] with worse breast cancer outcomes. In this study, both *MEK2* and *S100P* were upregulated in blood and tumor samples of invasive breast cancer. *S100P* was also upregulated in DCIS patient blood samples, indicating its potential clinical utility as a non-invasive biomarker for breast cancer diagnosis by itself. Together with the previous findings reported in the literature, this study shows that our identified signature genes are diagnostic and prognostic in breast cancer and are functionally involved in tumorigenesis and progression.

*PBX2* is also part of a 10-gene model to predict breast lesions identified in a whole-blood transcriptional profiling study [64]. *PBX1* and *PBX2* bind in a cooperative mechanism to DNA with both homeobox-containing genes *HOXB7* and *HOXB8* [65,66]. The introduction of *HOXB7* strongly increased the tumorigenic properties of breast cancer cells SkBr3 (SkBr3/B7) [67]. The functional requirement of the oncogenic activity of *HOXB7* was proven with a dominant-negative *PBX1* mutant, *PBX1NT* [67]. *HOXB7* was shown to regulate the expression of Hox cofactors by increasing *PBX2* and decreasing *PBX1* in SkBr3 cells [67]. Interestingly, *HOXB7* is functionally involved in tumor cell growth promotion through the direct transactivation of *FGF2* [67], which is also part of our 28-gene signature. *FGF2* induces breast cancer growth by activating and recruiting *ERα* and *PRBΔ4* isoforms to *MYC* regulatory sequences [68]. The FGF2/FGFR1 paracrine loop is functionally involved in the crosstalk between breast cancer cells and tumor stroma. The activation of FGF2/FGFR1 paracrine signaling triggers the expression of the connective tissue growth factor (*CTGF*), causing the migration and invasion of MDA-MB-231 cells [69]. The entire mechanism involving *FGF2, PBX1/PBX2,* and *HOXB7/HOXB8* in breast cancer tumorigenesis is not clear. In this study, *FGF2* mRNA was downregulated in DCIS and invasive breast cancer tumors compared with normal breast tissues but upregulated in blood samples from DCIS and invasive breast cancer patients versus healthy individuals. Underexpression of FGF2 mRNA and protein was also reported in malignant human breast tissues compared with non-malignant tissues [70]. The observed discrepancy of *FGF2* mRNA expression between the patients’ blood and tissues might be caused by its different functional involvements in breast cancer tissues and blood.

In this study, we examined the protein expression of 28 marker genes with available antibodies, including MCF2, RAD52, RAD50, PBX2, SMARCD2, IRF5, MCM2, and IGAH2, in Western blots. The protein expression of RAD50, RAD52, and PBX2 was confirmed in breast cancer cell lines MCF7, AU565, BT474, and MDA453 and immortalized breast epithelial cell line MCF10A [71] (Figure 6A). The results for the rest of the markers without any detectable protein expression in Western blots are not shown. In the future, we will further examine the protein expression of the marker genes in different breast cancer tumors to investigate their functions.

Based on the association of the 28-gene expression signature with BRCA patient survival, response to 18 NCCN-recommended drugs, and proliferation, we designed mechanisms of action to inhibit BRCA cell growth, survival hazard genes, and pan-resistant genes to discover new drugs or new indications of existing drugs for treating breast cancer. The candidate compounds screened from CMap were further selected based on their efficacy in inhibiting BRCA cell growth measured with IC_50_ and EC_50_. Among the six identified drugs, bosutinib is a small molecule functioning as a dual inhibitor of BCR-ABL and Src tyrosine kinase for treating chronic myeloid leukemia (CML) and advanced solid tumors [72], including estrogen-positive breast cancer [73]. Dasatinib is an oral inhibitor of multiple tyrosine kinases, including BCR-ABL, Src, c-KIT, PDGFR-α, PDGFR-β, and ephrin receptor kinases [74]. Dasatinib is effective for treating CML and Philadelphia chromosome-positive acute lymphoblastic leukemia (Ph+ ALL) [74]. A combination of paclitaxel and dasatinib showed some clinical activity in HER2-negative metastatic breast cancer in a phase II study [75]. Dasatinib induced sensitivity to c-Met inhibition in TNBR cells in a pre-clinical study [76]. Src inhibitor PP-1 suppresses the invasiveness of breast cancer cells [77]. PP-1 and PP-2 inhibit breast cancer proliferation [78]. Another Src inhibitor, saracatinib, is in clinical trials for treating multiple solid tumors, including breast cancer and lung cancer [79]. The VEGFR inhibitor ZM-306416 was discovered as an EGFR inhibitor [80]. We previously discovered it as a new drug for treating lung cancer [81]. In this study, it was discovered as a new drug for treating breast cancer that was not known before. Characteristics of BRCA patient responses to ZM-306416 are provided in Appendix A.

## 4. Materials and Methods

### 4.1. Patient Samples

The first cohort contained 185 breast tissue samples. Gene expression was quantified using the Affymetrix U133 Plus 2 Array (GEO accession number GSE10780) from Chen et al. [39]. These 185 samples include 143 histologically normal breast tissue samples and 42 invasive ductal carcinomas (IDC) tissue samples collected from 90 breast cancer patients who underwent a mastectomy. Each mastectomy specimen, where feasible, was prosected to produce an IDC and up to five successively derived, adjacent normal tissue samples from the ipsilateral breast or the four quadrants of the contralateral breast. All 143 histologically normal breast tissue samples were confirmed to be free of atypical ductal hyperplasia (ADH) and in situ or invasive breast carcinoma by one breast pathologist.

The second cohort contained 8 patient samples with gene expression quantified using the Affymetrix U133A microarray (GEO accession number GSE2429) from Poola et al. [40]. Among the 8 samples, 4 tissue samples were ADHC (from patients with ADH who had cancer concurrently, who had cancer before diagnosis of ADH, or who developed cancer subsequently) and 4 samples were from patients with ADH who had no prior history of breast cancer and had not developed breast cancer in five years after diagnosis.

The third cohort contained 77 snap-frozen breast tissue samples, including 3 ductal carcinomas in situ (DCIS), 60 invasive breast carcinomas, 2 sarcomas of the breast, and 12 normal breast tissues. These samples were obtained from the West Virginia University (WVU) Tissue Bank or Cooperative Human Tissue Network (CHTN) operated by the US National Cancer Institute. Tumor tissues were collected in surgical resections and were snap-frozen at −80 °C until RNA extraction. This cohort was used for quantitative RT-PCR validation of the biomarkers.

The fourth cohort contained 33 paraffin-embedded breast cancer tumor specimens used for immunohistochemistry assays. These samples were obtained from the WVU Tissue Bank. Histological preparations of tumor sections were examined by pathologists. This study was approved with an IRB exemption from WVU to use de-identified patient samples.

The fifth cohort contained 48 blood samples collected from patients seen at the Betty Puskar Breast Center of the WVU Cancer Institute/Mary Babb Randolph Cancer Center. The blood samples had been collected by the WVU Biospecimen Core and Tissue Bank. The blood samples were drawn from normal individuals (*n =* 7), or patients diagnosed with breast cancer (*n =* 41), including DCIS (*n* = 3) and invasive breast cancer (*n* = 38), with the patient’s consent. No patients received any chemotherapy or radiation within the 6 months prior to the blood draw (but they were eligible if on either Herceptin and/or anti-hormone therapy). Patient enrollment criteria are provided in Appendix A.

The sixth cohort contained data of log_2_-transformed Illumina HiSeq sequencing samples (*n =* 1093) and reverse-phase protein array (RPPA) expression normalized samples (*n =* 886) of The Cancer Genome Atlas on breast cancer (TCGA-BRCA). The datasets were retrieved from the online platform LinkedOmics [82] (http://www.linkedomics.org/, accessed on 10 March 2023).

### 4.2. Blood Collection

Blood for research was drawn under sterile conditions using the BD Vacutainer Safety-Lok Blood Collection Set and the One-Use Holder through venipuncture into PAXgene^®^ Blood RNA Tubes (Qiagen, Germantown, MD, USA) by a certified phlebotomist after the patient’s consent was obtained. PAXgene tubes were collected following the protocol to allow at least 10 s for a complete blood draw to take place or till the blood had stopped flowing into the tube (~2.5 mL of blood). After gently mixing by inverting the PAXgene Blood RNA Tubes 8 to 10 times, the blood samples were stored at 4 °C overnight.

### 4.3. RNA Extraction, Quality, and Concentration Assessment

Total RNA was extracted from 77 frozen breast tissue samples using the RNeasy mini kit according to the manufacturer’s protocol (Qiagen, Germantown, MD, USA). RNA was eluted in 30 μL of RNase-free water and stored at −80 °C. Total RNA was extracted from 48 blood samples. Blood samples were stored in PAXgene blood tubes (Qiagen, Germantown, MD, USA), which were equilibrated to room temperature for 2 h prior to isolation. Whole-blood RNA isolation was carried out using the PAXgene Blood miRNA Kit (Qiagen, Germantown, MD, USA) following the standard protocol. RNA was eluted with 40 μL Buffer BR5 directly onto the spin column membrane at 20,000 g twice. RNA was denatured by incubating for 5 min at 65 °C and then stored at –80 °C. The quality and integrity of the total RNA, 28S/18S ratio, and a visual image of the 28S and 18S bands were evaluated on the 2100 Bioanalyzer RNA 6000 Nano LabChip (Agilent Technologies, Santa Clara, CA, USA). The concentration of the RNA was assessed using the Nanodrop 1000 Spectrophotometer (NanoDrop Technologies, Wilmington, DE, USA).

### 4.4. Generation of Complementary DNA (cDNA)

Only high-quality RNA extracted from breast cancer samples was selected for conversion to cDNA. Reverse transcriptase polymerase chain reaction was used to convert single-stranded RNA to double-stranded cDNA using a Techne^®^ TC-312 PCR instrument (MIDSCI, St. Louis, MO, USA). For standardization across all samples, one microgram of RNA was used to generate cDNA.

### 4.5. Real-Time RT-PCR Low-Density Arrays

Two TaqMan Low-Density Array endogenous control gene cards (Applied Biosystems/Thermo Fisher Scientific Corporation, Waltham, MA, USA) were run on an Applied Biosystems PRISM 7900HT Sequence Detection System for 16 breast cancer tumor samples to identify genes that had the most relatively constant expression in different tissue samples. Four control genes, namely *18S, GAPDH, HMBS*, and *IPO8*, had constant expression in breast cancer tissue samples. Constant mRNA expression of *18S, HMBS*, and *IPO8* genes was also confirmed for all breast tissue samples using the individual TaqMan^®^ Gene Expression Assays. We used 384-well microfluidic low-density array plates designed to contain these 4 endogenous control genes and 26 breast cancer marker genes with 2 unknown genes removed from the original 28-gene signature. The primers on the plate are provided in Appendix A. The average expression of the four control genes was more stable than some single genes, but it was not the most stable compared to the best control gene in the analysis of breast cancer subtypes, stages, grades, or primary vs. recurrent samples. In the final analysis, *18S* was used as the control gene in the analysis of breast cancer samples vs. normal samples, and *HMBS* was used as the control genes of the analysis across breast cancer stages, grades, and primary vs. recurrent breast cancer samples.

The mRNA expression of the 26 signature genes was measured in each of the breast cancer tumors and normal breast tissues through RT-PCR using TaqMan^®^ Gene Expression Assays on a 7900 HT Fast RT-PCR instrument (Applied Biosystems/Thermo Fisher Scientific Corporation, Waltham, MA, USA). On each plate, 4 patient samples were loaded, and each primer was measured in triplicates. The report generated using Applied Biosystems SDS2.3 software included the number of cycles required to reach threshold fluorescence (Ct) and relative quantification (RQ), which numerically defined the expression pattern for the genes.

### 4.6. Statistical Analysis of Real-Time RT-PCR Data

∆Ct represents the normalized gene expression relative to the control gene HMBS in that sample. The average ∆Ct was calculated for each tumor type, including ductal carcinoma in situ, sarcoma, and invasive breast carcinoma. The fold change of each gene marker was computed for each tumor type vs. normal breast tissue samples. All statistical analysis was based on ∆Ct values, using two-tailed unpaired *t*-tests. *p* < 0.05 was considered statistically significant.

### 4.7. Stromal and Immune Infiltration

xCell [41] is a computational tool that predicts the relative abundance of immune and stroma cell types in complex tissue samples using gene expression data. The tool is based on a novel gene signature-based method that enables accurate quantification of 64 immune and stroma cell types, including T cells, B cells, natural killer cells, macrophages, fibroblasts, and endothelial cells. The xCell R package version 1.1.0 was used for the analysis in this study.

To analyze the immune microenvironment of patient samples, xCell scores were calculated using the single-sample gene set enrichment analysis (ssGSEA) method. High xCell scores indicate that the corresponding cell type is present in varying levels across the samples, whereas low xCell scores indicate that the cell type is present in similar levels across all the samples.

The association between gene expression and immune infiltration was determined using TIMER 2.0 [42,43], which is an updated version of the Tumor Immune Estimation Resource (TIMER). Accessible via its website at http://timer.cistrome.org/ (accessed on 19 March 2023), TIMER 2.0 is a valuable web server and database that offers researchers a comprehensive resource for investigating immune cell infiltrates in various types of cancer, including B cells, macrophages, myeloid dendritic cells, neutrophils, CD4^+^ T cells, and CD8^+^ T cells. The TCGA-BRCA cohort [82] was analyzed with xCell and TIMER2.0.

### 4.8. Protein Expression Validation Using Western Blot Analysis

Anti-PBX2 (sc-101853), -RAD50 (sc-20155), and -RAD52 (sc-8350) antibodies were obtained from Santa Cruz Biotechnology (Santa Cruz, CA, USA). Anti-MCF2 was obtained from Cell Signaling Technology (Danvers, MA, USA). Anti-IGHA2 was ordered from Abnova (Walnut, CA, USA). Anti-SMARCD2, -IRF5, and -MCM2 were ordered from AbCam (Cambridge, MA, USA). Anti-NEDD9 (HEF1) is a custom-made antibody [47]. β-Actin (13E5) Rabbit mAb #4970 was ordered from Cell Signaling Technology (Danvers, MA, USA). A protein extraction kit was ordered from *EMD* (Gibbstown, NJ, USA). Western blot analysis was performed on normal breast cells (MCF10A) and breast cancer cells (MDA-MB-231) according to the methods described previously [83]. Specifically, MCF10A and MDA-MB-231 cells were cultured in MEGM and DMEM, respectively, with 10% fetal bovine serum and 5% CO_2_ at 37 °C. The cells were lysed, and the lysates were subjected to SDS-PAGE, followed by immunoblotting with anti-PBX2, -RAD50, and -RAD52 antibodies, respectively. The remaining antibodies were not detected in the tested breast cancer cells in the Western blot analysis.

### 4.9. Protein Expression Validation Using Immunohistochemistry Analysis

Histologic slides containing tumor tissue from 33 patients were stained using the Ventana BenchMark Auto Stainer along with normal breast tissue from cancer patients, which was used as the control. The following is a brief outline of the protocol used on the auto-stainer: (1) histologic sections were deparaffinized and antigen retrieval was performed through incubation in Cell Conditioning (CC1) Solution for 30 min; (2) the primary antibody (rabbit polyclonal anti-MEK2; AbCam Inc Cambridge, MA, USA; Catalogue # ab28834) was applied at a dilution of 1:50 for 60 min; (3) the slides were incubated for 18 min with a second antibody; and (4) counterstained with hematoxylin. Likewise, anti-S100P (Proteintech Group Inc, Rosemont, IL, USA; Catalog # 11803-1-AP) and anti-FGF2 (Santa Cruz Biotechnology, Santa Cruz, CA, USA; Catalog # sc79) were used. Human colon cancer tissue was used as a positive control. The following protocol was used: (1) slides from the same 33 patients were deparaffinized and antigen retrieval was performed using CC1 for 30 min; (2) the slides were incubated with the primary antibody for 32 min at a dilution of 1:200 and the second antibody for 20 min; and (3) the slides were counterstained with hematoxylin. Anti-PBX2, -RAD50, and -RAD52 antibodies used in Western blots were also applied in immunohistochemistry assays of the same 33 patient samples. The protein expression scores in immunohistochemistry were quantified by a certified pathologist in the range of 0 to 4 as follows: 0 = no staining; 1 = equivocal staining; 2 = weak staining; 3 = moderate staining; 4 = strong staining.

### 4.10. Microarray Data Processing

Due to the differences in microarray platforms, 26 signature genes were found in GSE10780, and 25 signature genes were found in GSE2429. For the genes with multiple matching probes, the median expression of the duplicates was used to represent the gene expression in building the classifiers. Detailed genes used in these two datasets are provided in Appendix A. 

### 4.11. Construction of Molecular Classifiers

A regression algorithm with a model tree (M5P) as the base learner was used to build classifiers based on the 28-gene signature using the software WEKA 3.4 [84]. The model tree algorithm is based on the decision tree, in which each leaf stores a linear regression model that predicts the class value of instances that reach the leaf. Thus, predictions from the model tree are continuous numerical values. In this study, the classification of disease states was performed with a regression algorithm that constructed a model tree for each class, and instances were categorized into the class with the larger predicted value from the model tree.

### 4.12. Time-Dependent Receiver Operating Characteristics (ROC) Curves and Area under the ROC Curve (AUC)

To evaluate the predictive performance of the proposed survival gene signatures, we employed time-dependent ROC analysis for censored data and AUC (area under the ROC curve) as our criteria to assess the 5-year survival predictions [85]. The time-dependent sensitivity and specificity functions are defined as follows:$$\begin{array}{l} sensitivity(c,t)=P\{X>c|D(t)=1\}\\ specificity(c,t)=P\{X\leq c|D(t)=0\} \end{array}$$

The corresponding ROC(*t*) curve for any time *t* is defined as the plot of sensitivity (*c, t*) versus 1-specificity (*c, t*), with cutoff point *c* varying. *X* is the covariate and *D*(*t*) is the event indicator (here, death) at time *t*. The area under the curve, AUC(*t*), is defined as the area under the ROC(*t*) curve. The nearest neighbor estimator for the bivariate distribution function is used for estimating these conditional probabilities accounting for possible censoring [86]. AUC can be used as an accuracy measure of the diagnostic marker; the larger the AUC the better the prediction model. An AUC equal to 0.5 indicates no predictive power, while an AUC equal to 1 represents perfect predictive performance. The analysis was performed using the software package R version 4.3.1.

### 4.13. Clustering Analysis

The cluster analysis was performed by using the Heatplus package in R version 4.3.1. The Euclidean method was used to compute the distance and the agglomeration method was “average”. Dendrograms were plotted using within-gene-scaled ∆Ct values.

### 4.14. Proliferation Assays

In this study, we aimed to identify the proliferation determinants of breast tumor cells by analyzing the genome-scale CRISPR knockout and RNAi knockdown screening data of breast cancer cell lines. The CRISPR knockout screening results consisted of Achilles (Avana Cas9 library) [87,88] and Achilles combined with Sanger’s Project SCORE (KY Cas9 library) [89] screens. The 22Q4 data release of the DepMap portal (accessed on 2 March 2023 at https://depmap.org/portal/download/all/) provided the gene effect estimates for 48 breast cancer cell lines. Additionally, project Achilles (https://depmap.org/R2-D2/, accessed on 13 April 2023) provided whole-genome RNAi screening data [90] for CCLE cell lines. For our research, we used the genome-wide dependency scores of 34 breast cancer cell lines from RNAi screening.

Based on their significance to cell proliferation in each cell line, genes were categorized as essential and non-essential genes. In each cell line, the median of normalized dependence scores for common essential genes was –1, as compared to 0 for non-essential genes. We defined a significant effect of CRISPR-Cas9 knockout or RNAi knockdown as a normalized dependence score of less than –0.5 in this study.

### 4.15. Cancer Cell Line Encyclopedia (CCLE)

We extracted comprehensive genetic information for accessible human breast cancer epithelial cell lines from the Cancer Cell Line Encyclopedia (CCLE) [91] release DepMap Public 22Q4 (https://depmap.org/portal/download/all/, accessed on 2 March 2023). Specifically, we obtained RNA sequencing data for 63 breast cancer cell lines, 41,707 annotated and filtered mutations created with Mutect2, and 2979 fusions produced from RNAseq data. Detailed tumor subtype and PAM50 classification of each cell line are provided in Appendix A.

Additionally, from a project by the Gygi lab [92] (https://gygi.hms.harvard.edu/publications/ccle.html, accessed on 6 March 2023), we acquired proteome information for 31 breast cancer cell lines. Both the mRNA and proteomic data were log2-transformed, and the mean of protein expression was centered at 0.

### 4.16. Drug Sensitivity in CCLE

This study used drug sensitivity data of human breast cancer cell lines from various sources. The DepMap portal provided the secondary Profiling Relative Inhibition Simultaneously in Mixtures (PRISM) [93] repurposing dataset (PRISM repurposing 19Q4, accessed on 3 April 2023), which includes screening results of 1447 compounds in 22 human breast cancer cell lines. Additionally, we downloaded the Genomics of Drug Sensitivity in Cancer (GDSC1 and GDSC2) datasets [94,95,96] from the CancerRxGene website (https://www.cancerrxgene.org/downloads/bulk_download, accessed on 3 April 2023). GDSC1 contains screening results of 51 breast cancer cell lines and 345 compounds, while GDSC2 includes 50 breast cancer cell lines and 190 compounds screened.

The drug activity measurements in these datasets, including IC_50_, ln(IC_50_), EC_50_, and ln(EC_50_), were used to investigate drug sensitivity in this study. Based on measures of their drug activity, the breast cancer cell lines from the CCLE dataset were divided into three groups: sensitive, resistant, and intermediate, which have been described in our prior works [97,98].

### 4.17. CMap

Connectivity Map (CMap) [37,38] is a bioinformatics software package that uses gene expression data to identify potential drugs or small molecules that can modulate specific biological pathways or disease states. By comparing the gene expression signature of a particular disease or cellular state with the gene expression profiles in CMap, drugs or small molecules that could potentially reverse or mimic the disease or cellular state can be identified. In this study, the CMap online tool (https://clue.io/, accessed on 7 March 2023) was utilized to explore potential drug repositioning based on selected gene expression signatures. The results with a raw connectivity score >0.9 and a *p*-value < 0.05 were considered significant and can be further investigated to provide insights into drug mechanism of action and identify potential therapeutic options.

### 4.18. Statistics Methods

This study utilized RStudio (version 2023.03.0 Build 386) with R version 4.2.1 as the primary tool for statistical analysis. The significance of differential expression between the two groups was evaluated using a two-tailed, unpaired Student’s t-test. We employed the Kaplan–Meier method to generate survival curves and perform survival analysis and conducted log-rank tests to assess the difference in survival probability between different groups. The prognostic evaluation and risk score model building were performed with univariate and multivariate Cox regression analyses. The R packages “survival (version 3.5.3)” and “survminer (version 0.4.9)” were used. To determine the degree of a linear relationship between the two sample groups, we used Pearson’s correlation test. A result was considered significant if the *p*-value was less than 0.05.

## 5. Conclusions

This study showed that the 28-gene signature previously reported to be prognostic of breast cancer and ovarian cancer clinical outcome is also diagnostic of breast cancer and can classify invasive cancer tumors from normal and ADH tissues. This gene assay was evaluated in multiple independent patient cohorts in solid tissues. This gene signature can further stratify basal-like and luminal A BRCA patients into different prognostic groups with distinct immune cell activities. Using seven genes within this assay, breast cancer patients could be separated from normal and DCIS patients using peripheral whole-blood samples. These results show its feasibility to be used in diagnostic tests on biopsy or as a minimally invasive test on patient blood samples, upon further validation with larger patient cohorts and prospective evaluation in the future. The functional involvement of multiple signature genes in breast cancer tumorigenesis was also confirmed in protein assays. PBX2 and RAD52 protein expression levels in IHC are prognostic in invasive breast cancer patients. The mRNA and protein expression of multiple signature genes is associated with response to 18 NCCN-recommended drugs for treating breast cancer. Eleven signature genes had a significant effect on human BRCA cell lines in CRISPR-Cas9/RNAi screening. Based on this 28-gene expression signature, the VEGFR inhibitor ZM-306416 was discovered with new indications for treating breast cancer. The results presented in this study indicated that the 28-gene signature can be used for improving diagnosis, treatment selection, and drug discovery for breast cancer.

## 6. Patents

The 28-gene signature is under U.S. Non-Provisional Patent 8,030,060. The results in this manuscript are included in a provisional patent 63/509,532 filed by West Virginia University.

## Figures and Tables

**Figure 1 ijms-24-10561-f001:**
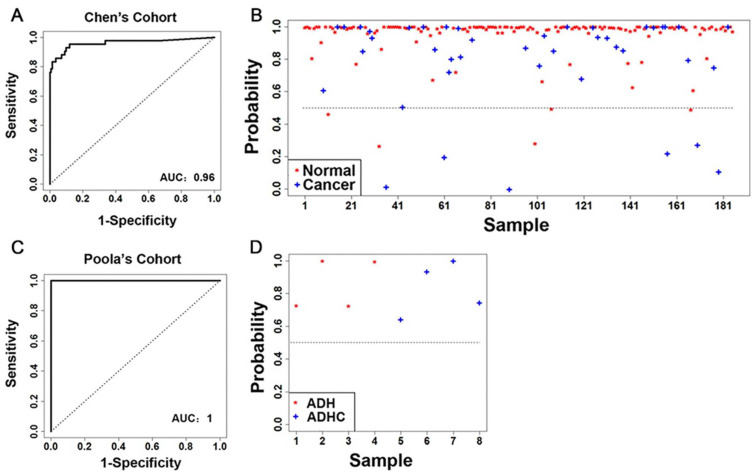
Accuracy of the 28-gene signature in prediction of breast cancer risk in two patient cohorts. (**A**) The ROC with AUC and (**B**) detailed performance in individual patients of the 28-gene signature in classifying invasive breast carcinoma from normal breast tissues in the cohort from Chen et al. [39] (*n* = 185). (**C**) The ROC with AUC and (**D**) detailed performance in individual patients of the 28-gene signature in classifying ADH from ADHC in the cohort from Poola et al. [40] (*n* = 8). ROC: time-dependent receiver operating characteristics curves. AUC: area under the ROC curve. ADH: atypical ductal hyperplasia. ADHC: atypical ductal hyperplasia with future cancer development.

**Figure 2 ijms-24-10561-f002:**
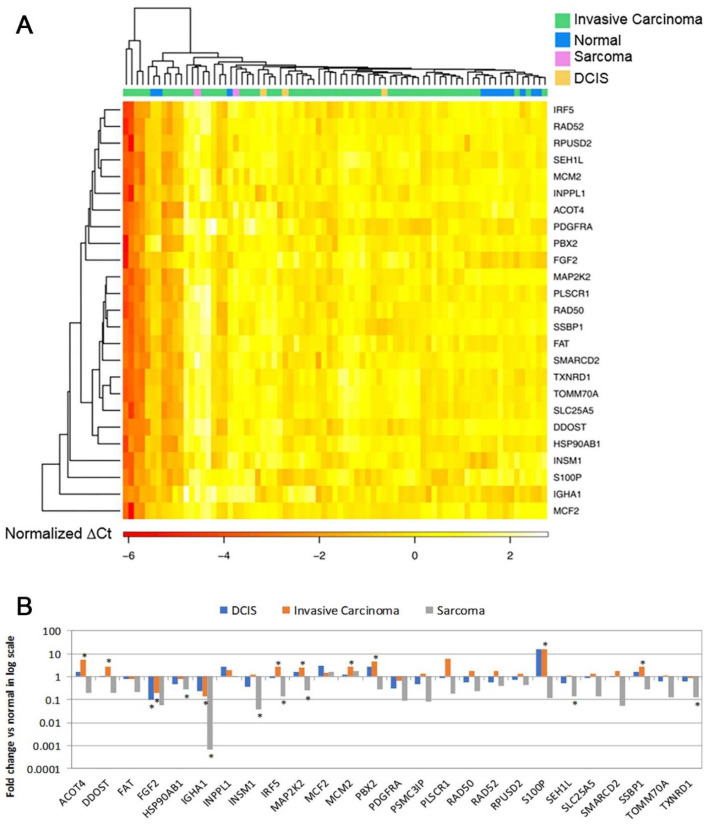
RT-PCR validation of the 28-gene signature in breast cancer tumors versus normal breast tissues (*n* = 77). (**A**). A heatmap of 25 genes separating normal breast tissues from invasive carcinoma, sarcoma, and DCIS. (**B**). Log_2_ scale fold change of each gene in each of the tumor types DCIS (*n* = 3), invasive carcinoma (*n* = 60), and sarcoma (*n* = 2) vs. normal breast tissues (*n* = 12), respectively. * *p* < 0.05 in two-tailed unpaired *t*-tests. DCIS: ductal carcinoma in situ.

**Figure 3 ijms-24-10561-f003:**
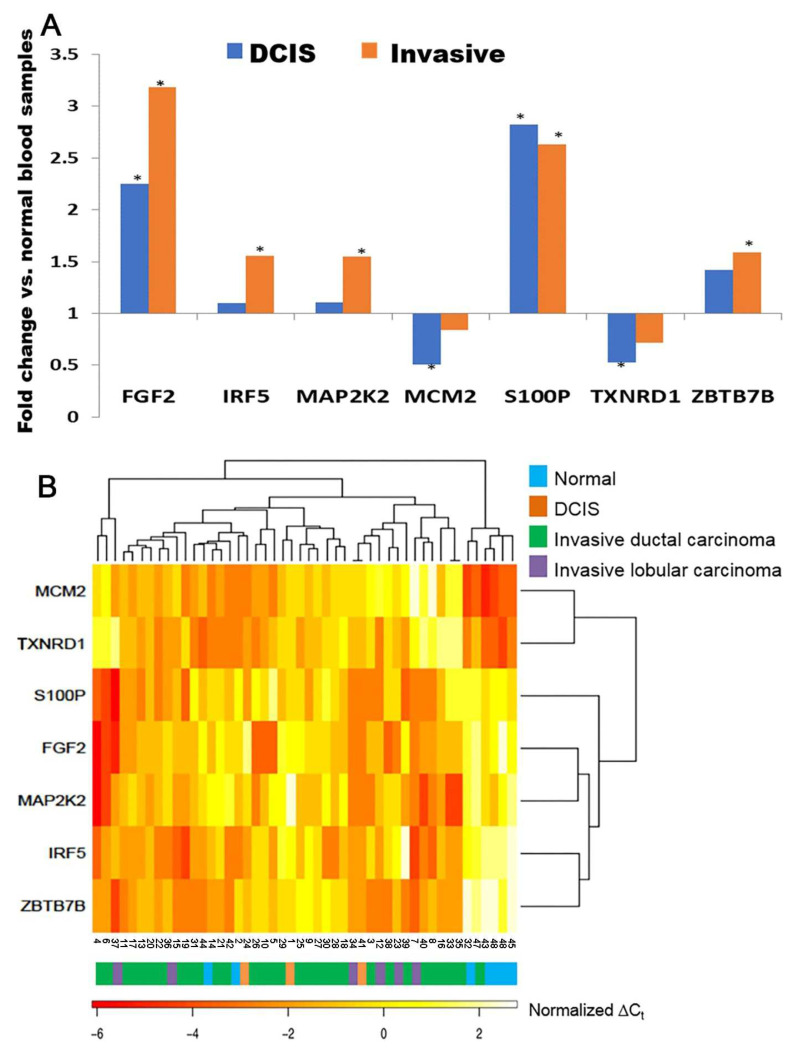
RT-PCR validation of the 28-gene signature in blood samples (*n* = 48). (**A**). Marker genes with significant differential expression in breast cancer patient blood samples DCIS (*n* = 3) or invasive carcinoma (*n* = 38) vs. normal blood samples (*n* = 7), respectively. * *p* < 0.05 in two-tailed unpaired *t*-tests. (**B**). A heatmap of the mRNA expression of marker genes in breast cancer blood samples and normal blood samples. DCIS: ductal carcinoma in situ.

**Figure 4 ijms-24-10561-f004:**
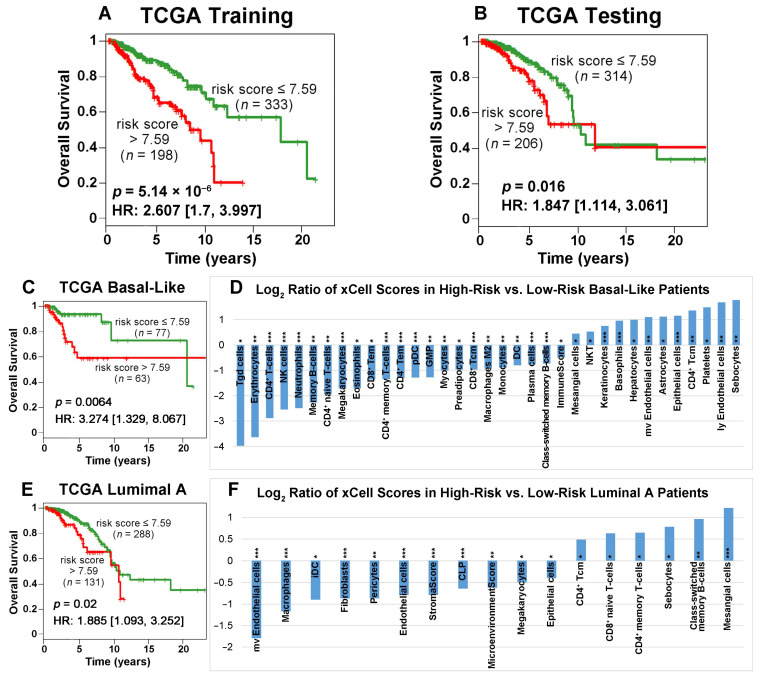
Kaplan–Meier analysis of 25 available genes in training (**A**) and testing (**B**) sets randomly partitioned from the TCGA-BRCA data. Patients were grouped by risk scores with a cutoff value of 7.59. Among the breast cancer subtypes in TCGA-BRCA, the low-risk patient group survived significantly longer than the high-risk group for basal-like (**C**) and luminal A (**E**) patients. The plots show patient survival outcomes in the first 10 years after surgery. The log_2_ ratios of xCell scores between the high-risk and low-risk groups of basal-like (**D**) and luminal A (**F**) patients are presented. Statistical analysis was performed using two-sample *t*-tests to assess the difference between the two groups (*: *p* < 0.05; **: *p* < 0.01; ***: *p* < 0.001).

**Figure 5 ijms-24-10561-f005:**
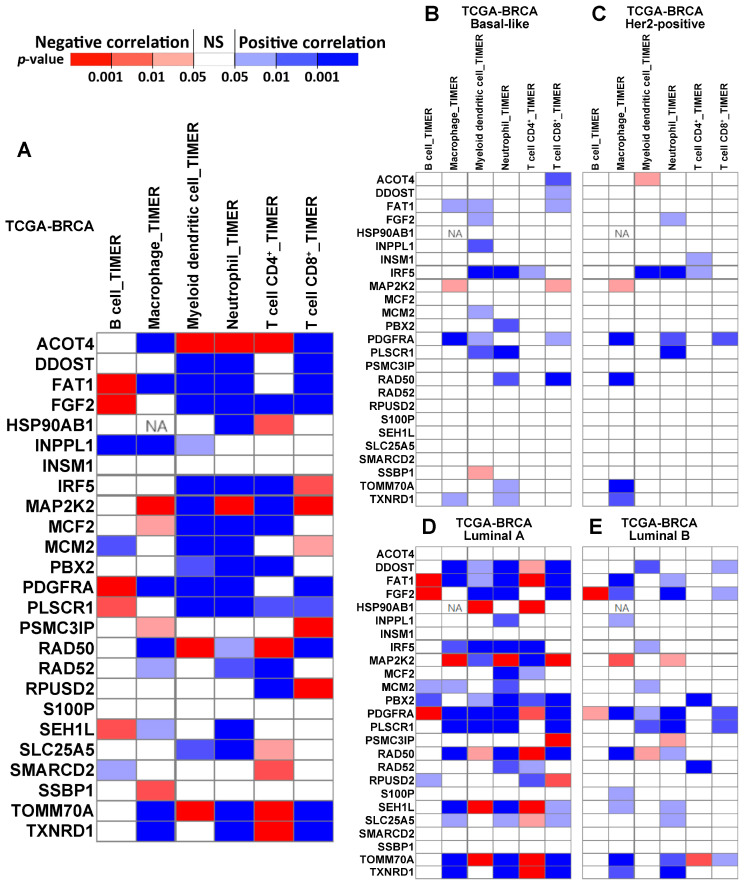
The correlation between the immune infiltration level and gene expression, estimated with TIMER, in different subtypes of breast cancer patients. Panel (**A**) shows the correlation in all TCGA-BRCA patients (*n* = 1100). Panels (**B**–**E**) show the correlation in specific subtypes of BRCA, including basal-like (*n* = 191), Her2-positive (*n* = 82), luminal A (*n* = 568), and luminal B (*n* = 219). ‘NA’ indicates that the data are not available.

**Figure 6 ijms-24-10561-f006:**
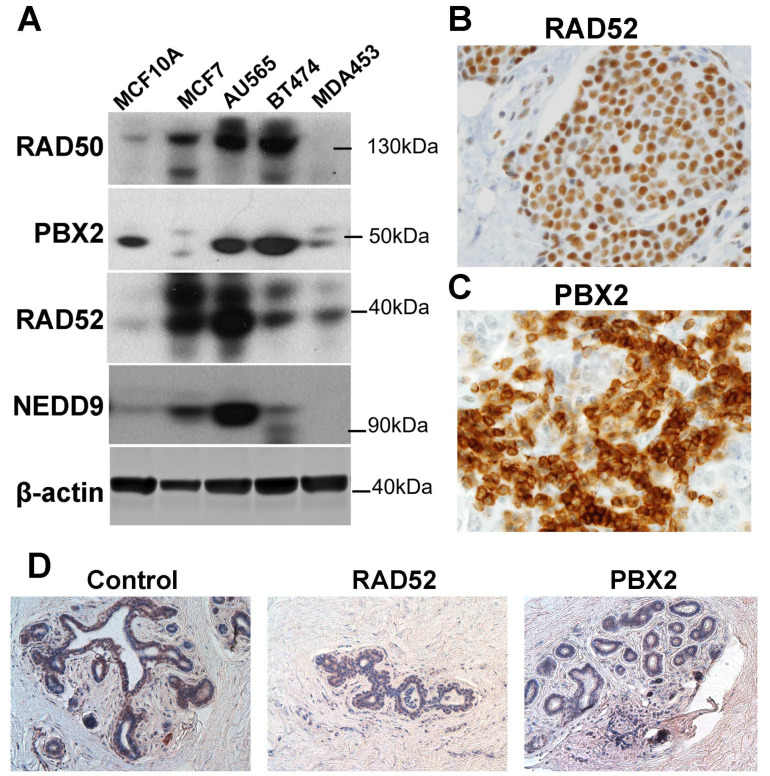
Protein expression validation of the marker genes. (**A**). Protein expression of PBX2, RAD50, and RAD52 in cell lines MCF10A, MCF7, AU565, BT474, and MDA453 in Western blots. (**B**). Immunohistochemistry (IHC) staining of RAD52 in breast cancer tumor tissue. (**C**). IHC staining of PBX2 in breast cancer tumor tissue. Both pictures (**B**,**C**) were shot with an objective of 40×. (**D**). IHC with control, RAD52, and PBX2 in normal breast tissue, viewed at 20×.

**Figure 7 ijms-24-10561-f007:**
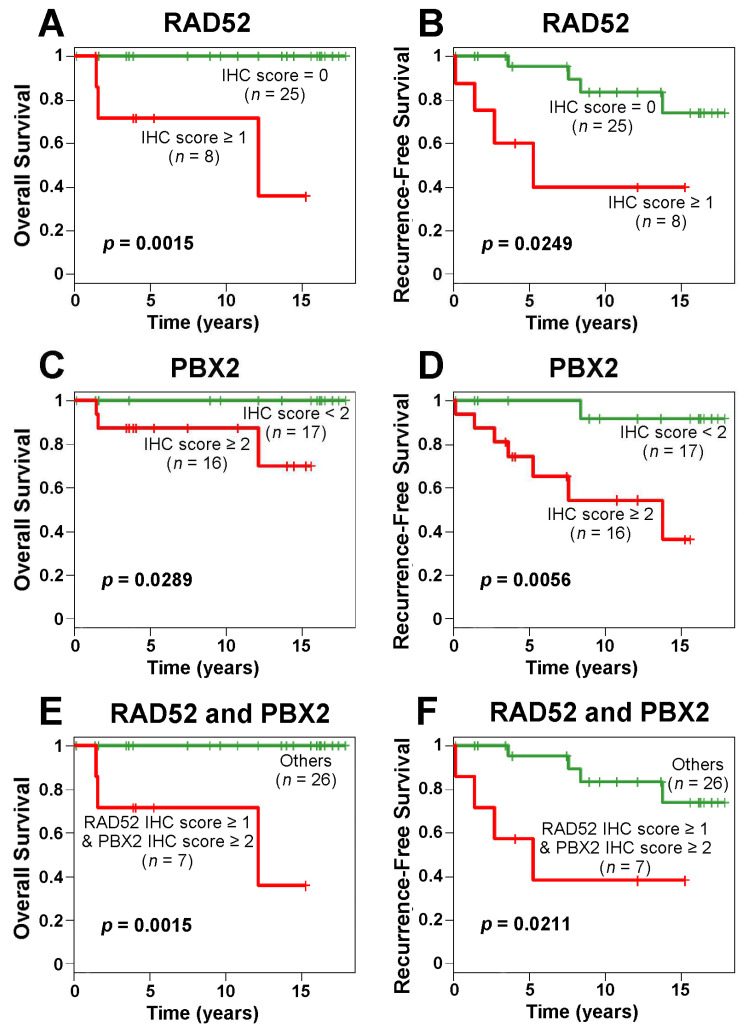
Kaplan–Meier curves of RAD52 and PBX2 protein expression in immunohistochemistry (IHC) for breast cancer overall survival and recurrence-free survival. IHC scores of RAD52 (**A**,**B**) and PBX2 (**C**,**D**) were used in patient stratification, respectively. The combined IHC scores of RAD52 and PBX2 were used to stratify patients into different prognostic groups in overall survival (**E**) and recurrence-free survival (**F**) analyses.

**Figure 8 ijms-24-10561-f008:**
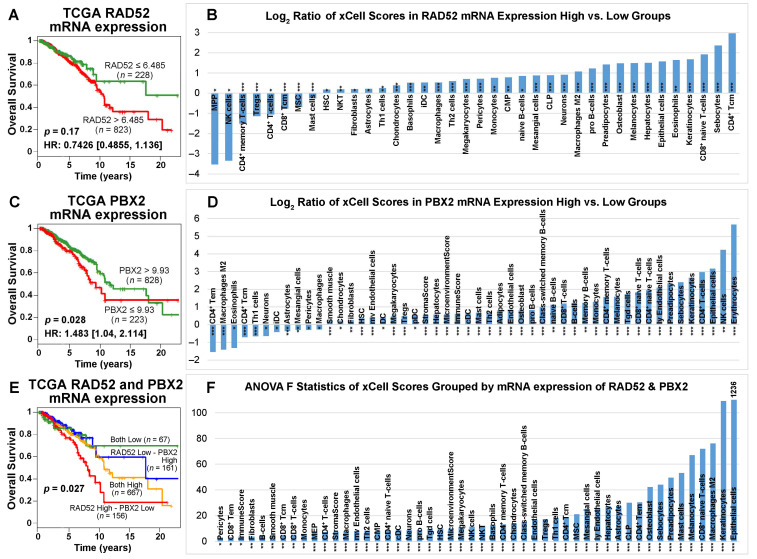
Patient survival analysis and immune cell type activities in patient tumors grouped by the expression level of *RAD52* and *PBX2* in TCGA-BRCA. (**A**) When *RAD52* mRNA was overexpressed, patients survived for a shorter length of time. Kaplan–Meier analysis was not significant. (**C**) When *PBX2* mRNA was overexpressed, patients survived significantly longer. (**E**) When the patients were grouped by the expression level of *RAD52* and *PBX2* together, the patient group with both genes expressed lower survived the longest among the four groups; the patient group with higher *RAD52* mRNA expression and lower *PBX2* mRNA expression survived the for the shortest length of time among the four groups. Panels (**B**,**D**) show the log_2_ ratio of xCell scores of patient samples grouped by the mRNA expression level of *RAD52* and *PBX2*, respectively. Cell types that had significantly different activities between the high- and low-expression groups in two-sample *t*-tests were included. Panel (**F**) shows the *F*-statistics of xCell scores grouped by the mRNA expression level of *RAD52* and *PBX2*. Cell types that were significant in the ANOVA tests were included. *: *p* < 0.05; **: *p* < 0.01; ***: *p* < 0.001.

**Figure 9 ijms-24-10561-f009:**
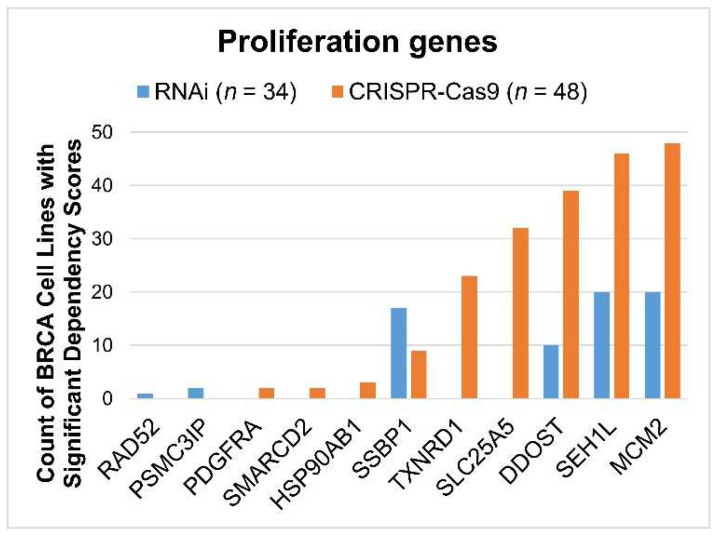
The proliferation of selected genes in human breast cancer (BRCA) cell lines as analyzed through CRISPR-Cas9 and RNAi screening. For each gene, the count of BRCA cell lines with significant dependency scores (<−0.5) is shown for CRISPR-Cas9 (*n* = 48) and RNAi (*n* = 34), respectively. Genes that did not have a significant impact on BRCA cell lines were not included in this figure.

**Figure 10 ijms-24-10561-f010:**
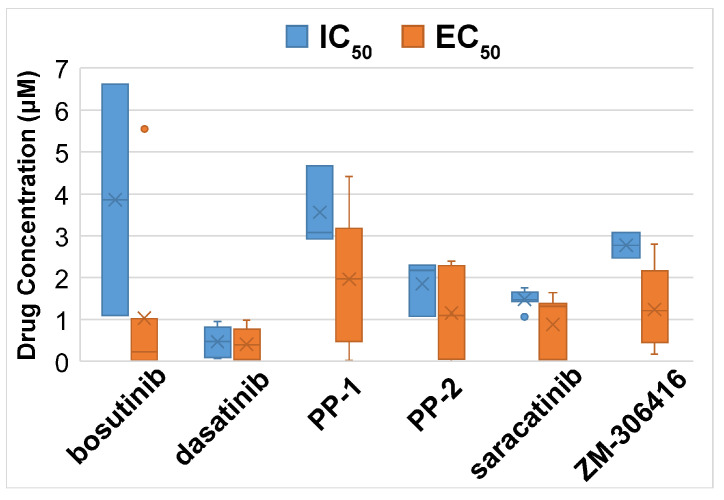
The average drug concentration of selected drugs in the CCLE human BRCA cell lines (*n* = 22). Dasatinib had a low average concentration of drug response across the cell lines tested. The measurements of EC_50_ = 4.49 × 10^7^ µM of dasatinib in cell line HCC38 and EC_50_ = 24.52 µM of saracatinib in cell line HCC1937 were outliers and removed from the plots.

**Table 1 ijms-24-10561-t001:** Resistant and sensitive genes to 18 drugs recommended in the NCCN Breast Cancer guidelines, as determined using mRNA and proteomics data from CCLE human BRCA cell lines. Genes with a significant association with mRNA expression are indicated in italic font, and genes with significant results for protein expression are indicated in normal font. Pan-resistant or pan-sensitive genes in terms of mRNA expression are indicated in bold font. Genes with concordant mRNA and protein expression association with drug response are highlighted in yellow. Genes with at least one non-silent mutation in at least one BRCA cell line are in blue; genes with fusions in BRCA cell lines are marked in orange; genes with both fusions and non-silent mutations in BRCA cell lines are in red.

Drug	Resistant	Sensitive
5-fluorouracil	***HSP90AB1**, **INPPL1**, TXNRD1,* FGF2	*DDOST*
alpelisib	FGF2, *PSMC3IP, **RAD50**, S100P*	***PLSCR1**, SEH1L, SMARCD2,* PBX2, S100P
capecitabine	***HSP90AB1**, MCM2*	
carboplatin		*MAP2K2*
cisplatin	*PDGFRA, PSMC3IP*	*DDOST, FGF2, **PLSCR1**, SLC25A5, SMARCD2,* MCM2
cyclophosphamide	FGF2, MCM2	*MCM2*
docetaxel	***INPPL1**, **IRF5**, RAD52, SMARCD2*	***RPUSD2**, S100P, SEH1L, SLC25A5*
doxorubicin		*PDGFRA, **PLSCR1**, PSMC3IP, **RPUSD2**, SEH1L, SLC25A5,* MCM2
fulvestrant	*TXNRD1,* FGF2, S100P	*ACOT4, FGF2, INSM1, MCM2, **RPUSD2**, S100P, SMARCD2*
gemcitabine	*ACOT4, FGF2, INSM1, SMARCD2*	*MAP2K2, PSMC3IP, S100P, SEH1L, SLC25A5, TXNRD1*
ixabepilone		*DDOST, **PBX2**, PDGFRA, **PLSCR1**, **RPUSD2***
lapatinib	***INPPL1**, INSM1, PSMC3IP, S100P, SSBP1*	*SMARCD2,* PBX2, S100P
methotrexate	***HSP90AB1**, MAP2K2,* PBX2, RAD52	
neratinib	*PSMC3IP, **RAD50**, S100P, SLC25A5, TXNRD1*	PBX2
olaparib	FGF2	*FGF2, MAP2K2, PDGFRA, RAD52,* S100P *, SEH1L, SLC25A5, SSBP1, TXNRD1,*
paclitaxel	*SMARCD2, SSBP1,* FGF2, MCM2	*SEH1L, SLC25A5*
tucatinib	*DDOST, PSMC3IP*	FGF2, PBX2, RAD52
vinorelbine	* **INPPL1** *	*DDOST, **PBX2**, **PLSCR1**, PSMC3IP, **RPUSD2**, S100P, SEH1L, SLC25A5, SSBP1, TXNRD1*

**Table 2 ijms-24-10561-t002:** Resistant and sensitive genes to dasatinib and PP-2 concordant in mRNA and proteomics data in CCLE human BRCA cell lines. Genes with at least one non-silent mutation in at least one BRCA cell line are in blue; genes with fusions in BRCA cell lines are marked in orange; genes with both fusions and non-silent mutations in BRCA cell lines are in red.

Drug	Resistant Genes	Sensitive Genes
dasatinib	ACHE, AP3B2, CLRN3, GDAP1, LOXL1, MTMR6, PLCB1, RAB6B, TIGAR	CNTN1, DHRS3, ERN2, GATA6, LRP1B, TDRP
PP-2		PALM

## Data Availability

All data are publicly available with access information provided in the manuscript and Appendix A.

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
