# Peer review of "Expression-Based Diagnosis, Treatment Selection, and Drug Development for Breast Cancer"

_ijms, 2023, doi:10.3390/ijms241310561_

Round 1

Reviewer 1 Report

Summary

-       The authors claim they have discovered a clinically applicable a 28 gene tests to detect precancerous normal tissue, premalignant lesions that will develop into invasive breast cancer.

-       The authors have demonstrated that some of the 28 genes are differentially expressed in invasive breast cancer samples (tumor or blood) compared to normal samples, and that these genes could help identify invasive breast cancer patients (diagnostic markers).

-       Risk scores computed for the expression of some of the 28 genes are able to inform on patients’ prognosis, specifically for Basal-Like and Luminal A subtypes. (Prognostic biomarkers) 

-       Patients with high or low risk score have a different immune cell profile.

-       The authors further investigated 2 genes out of the 28 gene-signature: RAD52 and PBX2.

-       The authors investigated whether or not the genes had an impact on cell growth in vitro from public data and also drug options.  

Introduction

Line 67: “TNBC or basal-like” -> These two terms encompass ER-/PR-/HER2- BC, but it is not a synonym, hence the authors should say and instead of or. The techniques to attribute TNBC (IHC) or basal-like subtype (PAM50) are different. This is very important.

Line 118: Please define NCCN.

Results

2.1 28-gene signature as a diagnostic assay for breast cancer

Line 122 and 123: According to Materials and Methods section only 26 and 25 genes of the 28-gene signature were used in this analysis. Please modify accordingly where it is written 28-gene if not all 28 genes were used.

Line 132: It is not clear in the methods (for the cohort #1) that there were normal tissue patients and breast cancer patients. The authors should either modify this sentence to represent what is written in the materials section or modify the materials section to specify that there are normal patients and breast cancer patients.

2.2 RT-PCR validation of the 28-gene signature in breast cancer tissue samples

Line 145: Should read “26 genes out of the 28-gene signature” to truly reflect the results presented.

Line 166: Only 25 genes are presented in the heatmap. Figure title should be modified to accurately represent what is shown in the figure.

Line 167: Please write the number of samples per tumor type.  

Supplementary File Si Figure S2-S5

-       Please define “HMBS plate wise”, either in the figure title or in the main text.

-       Fig S4: Who is Jill?

-       Fig S2 to S5: Figure legend should be changed to properly understand the figure.

2.3 RT-PCR validation of 28-gene signature in breast cancer blood samples

Line 173: It is not clear in the methods section that this cohort included DCIS and invasive breast cancers. Please see comment for line 585.

Line 182: It would be more coherent to talk about protein expression of the tumor samples in prior section 2.2, as the same tumors are indirectly presented in this section (2.2), specifically in Figure S5.

Line 190: Please specify number of samples per tumor types.  

2.4 Prognosis within breast cancer subtypes using next-generation sequencing data

Line 195: Do these randomly split groups have similar proportions of PAM50 subtypes? Please mention. The TCGA Testing group has similar curves to TCGA luminal A, could it be because this group includes more samples of this particular subtype?

Line 197: What are the 25 genes available? Please mention.

Line 202/ Figure 4 B:

-       Is there any information further than 10 years, as it seems that the patients with a risk-score less than or equal to 7.59 are under the curve of the other risk group after 10 years.

Line 213-214: Number of patients per subtype?

2.6. Protein expression validation of the signature genes  

Line 287: There is no control protein for Western blots? Please add the blot to the figure.

Line 297: Figure 7 title and legend is not informative enough. Please add details.

Line 323: Please add source regarding HIV.

2.9. Discovery of new drugs with CMap

Line 400: Please define IC50, EC50.   

Discussion

Throughout the Discussion section: Whenever all 28 genes were not used in the analysis, refrain from using “28-gene signature” and specify the number of genes that were truly used.

Line 459-462: With the gene signature, the authors did not identify patients with high probability of developing invasive breast cancer that already had DCIS, but they have shown differential expression between normal tissue, DCIS and invasive cancer.

Therefore, the sentence should be changed accordingly.

Line 480: No functional assays were performed. Please remove functional.  

Line 520-522: Why not mentioned all the antibodies used in the method? Please add these details in the appropriate section.

Materials and Methods

4.1 Patient samples

Line 558: The authors have written « 185 samples include 143 histologically normal breast tissues and 42 invasive ductal carcinomas (IDC) tissues collected from 90 breast cancer patients ». Are there breast cancer patients that are not included in IDC samples, and do we have to understand that there are around 2 samples of normal tissue per breast cancer patient?

Line 561: Do the authors mean Ductal hyperplasia? If not, please modify ADH for the write acronym.

Line 576: No normal tissue samples in this cohort?

Line 584-585: “patients diagnosed with breast cancer (n=41)”. In the results regarding blood samples the authors talk about invasive breast cancer and DCIS. Hence the information about the number of patients and the type of breast cancer should appear in this paragraph.

4.5. Real-time RT-PCR low-density arrays and 4.6 Statistical analysis of Real-time RT-PCR data

Line 633 and line 643: Since four control genes were performed, why not use the mean of all endogenous control for relative quantification, instead of using only one gene? Please specify.

4.7. Stromal and Immune Infiltration

The authors should mention cohorts that are included in this analysis.

4.8. Protein Expression validation using western blot analysis

What antibody was used for control? Please specify

Mycoplasma tested cell lines? Please specify

4.10 Microarray Data processing

Line 697: What are the 26 genes? Please specify

Line 698: What are the 25 genes?  Please specify

4.15. Cancer Cell Line Encyclopedia (CCLE)

The authors should mention the proportion of each breast cancer subtype in breast cancer cell lines data used.  

Line 752: Please specify if the cell lines are breast cancer cell lines, and which ones.

Conclusion

Line 789: Authors did not demonstrate that the gene signature is able to classify DCIS, they only presented genes that were similarly or not expressed with invasive breast cancer compared to normal tissue or blood. No diagnostic assay was done. Hence the whole conclusion should be toned down.

Author Response

Reviewer 1

Reviewer: Summary

-       The authors claim they have discovered a clinically applicable a 28 gene tests to detect precancerous normal tissue, premalignant lesions that will develop into invasive breast cancer.

-       The authors have demonstrated that some of the 28 genes are differentially expressed in invasive breast cancer samples (tumor or blood) compared to normal samples, and that these genes could help identify invasive breast cancer patients (diagnostic markers).

-       Risk scores computed for the expression of some of the 28 genes are able to inform on patients’ prognosis, specifically for Basal-Like and Luminal A subtypes. (Prognostic biomarkers) 

-       Patients with high or low risk score have a different immune cell profile.

-       The authors further investigated 2 genes out of the 28 gene-signature: RAD52 and PBX2.

-       The authors investigated whether or not the genes had an impact on cell growth in vitro from public data and also drug options.  

Authors: We thank the reviewer for the thorough review and constructive comments.

Introduction

Reviewer: Line 67: “TNBC or basal-like” -> These two terms encompass ER-/PR-/HER2- BC, but it is not a synonym, hence the authors should say and instead of or. The techniques to attribute TNBC (IHC) or basal-like subtype (PAM50) are different. This is very important.

Authors: We corrected it.

Reviewer: Line 118: Please define NCCN.

Authors: It is now defined in the manuscript.

Results

2.1 28-gene signature as a diagnostic assay for breast cancer

Reviewer: Line 122 and 123: According to Materials and Methods section only 26 and 25 genes of the 28-gene signature were used in this analysis. Please modify accordingly where it is written 28-gene if not all 28 genes were used.

Authors: It is now clarified in lines 132 and 135, respectively.

Reviewer: Line 132: It is not clear in the methods (for the cohort #1) that there were normal tissue patients and breast cancer patients. The authors should either modify this sentence to represent what is written in the materials section or modify the materials section to specify that there are normal patients and breast cancer patients.

Authors: It is normal breast tissues vs. invasive breast carcinoma samples. It is now clarified in lines 133-134.

2.2 RT-PCR validation of the 28-gene signature in breast cancer tissue samples

Reviewer: Line 145: Should read “26 genes out of the 28-gene signature” to truly reflect the results presented.

Authors: Agree. It is now corrected.

Reviewer: Line 166: Only 25 genes are presented in the heatmap. Figure title should be modified to accurately represent what is shown in the figure.

Authors: It is now clarified in the figure legend (line 168).

Reviewer: Line 167: Please write the number of samples per tumor type.  

Authors: The sample size is now provided.

Reviewer: Supplementary File Si Figure S2-S5

-       Please define “HMBS plate wise”, either in the figure title or in the main text.

-       Fig S4: Who is Jill?

-       Fig S2 to S5: Figure legend should be changed to properly understand the figure.

Authors: All the figure legends are now revised.

2.3 RT-PCR validation of 28-gene signature in breast cancer blood samples

Reviewer: Line 173: It is not clear in the methods section that this cohort included DCIS and invasive breast cancers. Please see comment for line 585.

Authors: The breast cancer samples include DCIS (n = 3) and invasive breast cancer (n = 38). It is now added in Materials and Methods in line 600.

Reviewer: Line 182: It would be more coherent to talk about protein expression of the tumor samples in prior section 2.2, as the same tumors are indirectly presented in this section (2.2), specifically in Figure S5.

Authors: We thank the reviewer for the comment. The protein expression results are now added in Section 2.2.

Reviewer: Line 190: Please specify number of samples per tumor types.  

Authors: The sample size is now added.

2.4 Prognosis within breast cancer subtypes using next-generation sequencing data

Reviewer: Line 195: Do these randomly split groups have similar proportions of PAM50 subtypes? Please mention. The TCGA Testing group has similar curves to TCGA luminal A, could it be because this group includes more samples of this particular subtype?

Authors: During the random split, we tried to keep the same proportion of samples in the training set (n=547) and the testing set (n=548). However, the specific number of each PAM50 subtype in the training set and the testing set will vary slightly in the random split.  In the TCGA breast cancer patient cohort, Luminal A has the largest number of patients (419 out of 1,051) in the survival analysis. That might be the reason our classifier produced similar results on Luminal A and the testing set.  It is now added to lines 225 to 231.

Reviewer: Line 197: What are the 25 genes available? Please mention.

Authors: The 25 available genes are provided in Supplementary File 3.

Reviewer: Line 202/ Figure 4 B:

-       Is there any information further than 10 years, as it seems that the patients with a risk-score less than or equal to 7.59 are under the curve of the other risk group after 10 years.

Authors: The Figure 4 is replaced with a new figure showing all the data. The patients with a risk-score less than or equal to 7.59 are under the curve of the other risk group after 10 years, which could be due to other factors such as loss of follow up and comorbidities, etc.

Reviewer: Line 213-214: Number of patients per subtype?

Authors: The sample size for each subtype in the survival analysis is now provided.

2.6. Protein expression validation of the signature genes  

Reviewer: Line 287: There is no control protein for Western blots? Please add the blot to the figure.

Authors: We tested whether the proteins were expressed in breast cancer cells using specific antibodies in Western Blots. We did not perform the Western Blots to evaluate the up or down regulation of these proteins. The protein expression levels were further assessed in patient tumors using IHC. So, no control proteins were used in the Western Blots.

Reviewer: Line 297: Figure 7 title and legend is not informative enough. Please add details.

Authors: Details are now provided.

Reviewer: Line 323: Please add source regarding HIV.

Authors: The reference is now added.

2.9. Discovery of new drugs with CMap

Reviewer: Line 400: Please define IC50, EC50.   

Authors: The definitions are now added.

Discussion

Reviewer: Throughout the Discussion section: Whenever all 28 genes were not used in the analysis, refrain from using “28-gene signature” and specify the number of genes that were truly used.

Authors: Corrected.

Reviewer: Line 459-462: With the gene signature, the authors did not identify patients with high probability of developing invasive breast cancer that already had DCIS, but they have shown differential expression between normal tissue, DCIS and invasive cancer.

Therefore, the sentence should be changed accordingly.

Authors: We agree with this comment. This section is changed accordingly.

Reviewer: Line 480: No functional assays were performed. Please remove functional.  

Authors: “functional” is removed from the sentence.

Reviewer: Line 520-522: Why not mentioned all the antibodies used in the method? Please add these details in the appropriate section.

Authors: The antibodies were added in Materials and Methods Section 4.8.

Materials and Methods

4.1 Patient samples

Reviewer: Line 558: The authors have written « 185 samples include 143 histologically normal breast tissues and 42 invasive ductal carcinomas (IDC) tissues collected from 90 breast cancer patients ». Are there breast cancer patients that are not included in IDC samples, and do we have to understand that there are around 2 samples of normal tissue per breast cancer patient?

Authors: Each mastectomy specimen, where feasible, was prosected to produce an IDC and up to five successively derived, adjacent normal tissue samples from the ipsilateral breast or the four quadrants of the contralateral breast. It is now added to the Materials and Methods section 4.1.

Reviewer: Line 561: Do the authors mean Ductal hyperplasia? If not, please modify ADH for the write acronym.

Authors: It is atypical ductal hyperplasia (ADH). It is now corrected.

Reviewer: Line 576: No normal tissue samples in this cohort?

Authors: That is correct. Only tumors were collected in this cohort for IHC assays.

Reviewer: Line 584-585: “patients diagnosed with breast cancer (n=41)”. In the results regarding blood samples the authors talk about invasive breast cancer and DCIS. Hence the information about the number of patients and the type of breast cancer should appear in this paragraph.

Authors: The subtype sample size is now added to line 600.

4.5. Real-time RT-PCR low-density arrays and 4.6 Statistical analysis of Real-time RT-PCR data

Reviewer: Line 633 and line 643: Since four control genes were performed, why not use the mean of all endogenous control for relative quantification, instead of using only one gene? Please specify.

Authors: Our statistical analysis showed that the mRNA expression of every control gene varied differently according to breast cancer subtypes, stages, grade, and primary vs. recurrent samples. The average expression of four control genes was more stable than some single genes, but it was not the most stable compared to the best control gene in the analysis of breast cancer subtypes, stages, grades, or primary vs. recurrent samples. In the final analysis, 18S was used as the control gene in the analysis of breast cancer samples vs. normal samples, and HMBS was used as the control genes of the analysis across breast cancer stages, grades, and primary vs. recurrent breast cancer samples.  It is now specified in lines 647-653.

4.7. Stromal and Immune Infiltration

Reviewer: The authors should mention cohorts that are included in this analysis.

Authors: TCGA-BRCA cohort was analyzed with xCell and TIMER2.0. It is now added to this section.

4.8. Protein Expression validation using western blot analysis

Reviewer: What antibody was used for control? Please specify

Authors: The Western Blots were performed to confirm protein expression prior to IHC assessment. SO no control antibody was used in Western Blots.

Reviewer: Mycoplasma tested cell lines? Please specify

Authors: We did not perform mycoplasma testing.

4.10 Microarray Data processing

Reviewer: Line 697: What are the 26 genes? Please specify

Line 698: What are the 25 genes?  Please specify

Authors: It is now specified in the manuscript. Detailed genes used in these two datasets are now provided in Supplementary File 3.

4.15. Cancer Cell Line Encyclopedia (CCLE)

Reviewer: The authors should mention the proportion of each breast cancer subtype in breast cancer cell lines data used.  

Line 752: Please specify if the cell lines are breast cancer cell lines, and which ones.

Authors: The cell lines are breast cancer cell lines. Detailed tumor type and PAM50 classification for each cell line are now added in Supplementary File S5.

Conclusion

Reviewer: Line 789: Authors did not demonstrate that the gene signature is able to classify DCIS, they only presented genes that were similarly or not expressed with invasive breast cancer compared to normal tissue or blood. No diagnostic assay was done. Hence the whole conclusion should be toned down.

Authors: We removed DCIS from this sentence.

Reviewer 2 Report

This work is presented as well structured and with consolidated and well demonstrated results. I find that the discovery of new biomarkers for cancer diagnosis, as well as in other pathologies, is of fundamental importance to improve good clinical practice.

I thank and congratulate the authors for their work.

Author Response

We thank the reviewer for the positive comments!

Round 2

Reviewer 1 Report

The authors have addressed most of the comments raised and have amended the manuscript accordingly. However, one important comment has not been addressed: the lack of controls for the western blots and the IHC. 

The authors should provide a control for the western blots or for the IHC, as they try to validate the expression seen in RT-PCR (normal vs breast cancer) in protein expression. 

Author Response

We thank the reviewer for raising this comment. We have retrieved the Western Blots and IHC results to get the relevant control data. The Western  Blots results have been revised substantially in Figure 6A and in the manuscript. Section 2.6 has been revised accordingly. The raw Western Blots images are provided for review. In IHC, normal breast tissue was used as a control. Human colon cancer tumor tissue was used as a positive control. The IHC control results in normal breast tissue are now provided in Figure 6D.